# Mitigation of Chromium Poisoning of Ferritic Interconnect from Annealed Spinel of CuFe$_2$O$_4$

## Muhammad Aqib Hassan *[ID] and Othman Bin Mamat

Department of Mechanical Engineering, Universiti Teknologi PETRONAS, 32610 Seri Iskandar,
Perak Darul Ridzuan, Malaysia; drothman_mamat@utp.edu.my
* Correspondence: aqibhassan71@gmail.com; Tel.: +60-10-257-5308

**Abstract:** Low-temperature solid oxide fuel cells permit the possibility of metallic interconnects over conventional ceramic interconnects. Among various metallic interconnects, the ferritic interconnects are the most promising. However, chromium poisoning in them adversely affects their performance. To resolve this issue, various coatings and pretreatment methods have been studied. Herein, this article encloses the coating of CuFe$_2$O$_4$ spinel over two prominent ferritic interconnects (Crofer 22 APU and SUS 430). The CuFe$_2$O$_4$ spinel layer coating has been developed by the dip-coating of both samples in CuFe$_2$O$_4$ slurry, followed by heat treatment at 800 °C in a reducing environment (5% hydrogen and 95% nitrogen). Additionally, both samples were annealed to further enhance their spinel coating structure. The morphological and crystallinity analysis confirmed that the spinel coating formed multiple layers of protection while annealing further reduced the thickness and improved the densities. Moreover, the area-specific resistance (ASR) and weight gain rate (WGR) of both samples before and after annealing was calculated using mathematical modeling, which matches with the experimental data. It has been noted that CuFe$_2$O$_4$ spinel coating improved the ASR and WGR of both samples which were further improved after annealing. This research reveals that the CuFe$_2$O$_4$ spinel is the promising protective layer for ferritic interconnects and annealing is the better processing technique for achieving the preferred properties.

**Keywords:** Solid Oxide Fuel Cell; copper-ferrous oxide; coating; metallic interconnect; SUS 430; Crofer 22 APU

## 1. Introduction

Limited reserves of fossil fuels and increasing demand for clean energy have increased the need for alternative sources of power. Among different power generation systems, the fuel cell is the most promising technology [1,2] that generates electricity from chemical reactions in an eco-friendly way.

Solid Oxide Fuel Cell (SOFC) is the most mature technology of fuel cells that mainly consists of two porous electrodes and a dense electrolyte [3]. The arrangement of a cell's components is shown in Figure 1 [4].

These cells are connected by using interconnects to increase power density. Modern technology of SOFC allows for the use of metallic interconnects instead of ceramics because metals have better electrical conduction at low temperatures, a compatible value of Co-efficient of Thermal Expansion (CTE), structural stability and high density [5–14]. Metallic interconnects include ferritic steel for better corrosion resistance. But, a limitation associated with the ferritic is that the chromium leaches out from the surface and penetrates into the cathode side, which reduces electrical conduction [14,15]. It is

noticed that the chrome layer ($Cr_2O_3$) oxidizes during operation and produces a volatile specie of chrome, i.e., $CrO_2(OH)_2$, as per the following reaction:

$$2Cr_2O_3 + 4H_2O + 3O_2 \rightarrow 4CrO_2(OH)_2$$

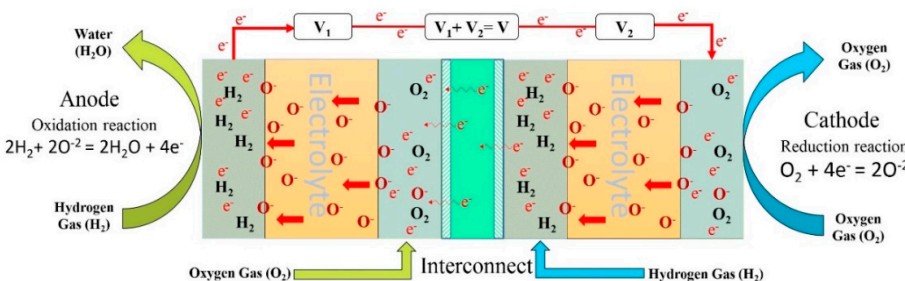

**Figure 1.** Block diagram of Solid Oxide Fuel Cell and gasses flow.

The penetration power of chromium hydroxide disturbs cathode performance which ultimately diminishes the power density of stack, which is generally known as "cathode poisoning". For the problem, different metallic interconnects have been fabricated for the reduction of chromium mitigation, and Crofer 22 APU has been specially designed for power collection of fuel cells. It is the ferritic steel that gives significant resistance in chromium migration and exhibits better properties [16–24]. At the same time, SUS 430 is commercial ferritic steel, which shows comparable results against operational hours of SOFC [15,25,26]. Therefore, it has recently come under research as a future metallic interconnect.

Despite the high performances of these alloys, interconnects require further reduction in oxidation [27,28], which raises the concept of external coating. It includes multiple compositions of spinel layers such as coatings of cobalt (Co), manganese (Mn), copper (Cu) and different reactive materials. Some of the Cu based spinel that contain Mn and Co are $CuMn_2O_4$, $CuMn_{1.8}O_4$, $Cu_{1.3}Mn_{1.7}O_4$, $CuMn_2O_4$, $Mn_{1.4}Co_{1.4}Cu_{0.2}O_4$ and $Cu_{0.3}Mn_{1.35}Co_{1.35}O_4$. The previous study states that the conductive nature of Cu provides high electrical conduction, but excessive addition of copper disturbs the thermal coefficient of the spinel [29,30]. At the same time, the spinel of $CuFe_2O_4$ provides good CTE compatibility with the base materials. Due to similarity the index in the chemical composition of iron between spinel and base material, spinel has good adhesion strength so that the layer does not spall out during the fabrication process and operational hours of SOFC. The spinel of $CuFe_2O_4$ gives better electrical conduction (5.2–7.6 S·cm$^{-1}$) [31]. Yue Pan et al. experimented with the performance of copper ferrous oxide in which the spinel was applied by magnetron sputtering method over bare Crofer 22 APU and preoxidized steel. Scanning Elecrton Microscope (SEM) and Energy Dispersive X-ray (EDX) results showed that the spinel formed a triple oxide layer on both steels. Preoxidation successfully reduced the internal formation $Cr_2O_3$ layer which ultimately decreased oxidation of chromium in the spinel. The enhancement in oxidation resistance decreases the area-specific resistance (ASR) by 38% after oxidation of 2520 h at 800 °C after formation of the triple oxide layer. It was found that the coating effectively resisted oxidation as the parabolic rate declined by 8% [32]. Other researchers, Shujiang Geng et al. thermally grew $CuFe_2O_4$ spinel over Crofer 22 APU by magnetron sputtering method. They found that the double layers were formed over steel substrate. Based on EDX and X-Ray Diffraction (XRD) results, the layers were identified as inner $Cr_2O_3$ and outer $CuFe_2O_4$ spinel layer. The top spinel layer effectively reduced Cr-migration and also lowered the weight gain rate of the spinel. Decreasing oxidation of the spinel dropped the ASR value by 14.3 m$\Omega$·cm$^2$ after oxidation 600 h at 800 °C [31].

This study examines the annealing effect over oxidation resistance and area specific resistance of Crofer 22 APU and SUS 430. Herein, the Cu-based spinel ($CuFe_2O_4$) was layered over ferrites by using the Dip Coating method. Coated samples were heated at 800 °C in tube furnace and later annealed in a box furnace. It is expected that the spinel will improve oxidation resistance of both

steels significantly, and copper will provide better conduction to the spinel. In addition, annealing will improve crystallinity and lattice structure of the spinel, which will assist in densification and reduction in porosity of the spinel layer.

## 2. Experimental

### 2.1. Materials

All the samples of JIS-SUS430 were collected by Trusted Asia Pacific Steel Supplier (E Steel Sdn. Bhd., 42100 Klang, Selangor, Malaysia), while Crofer 22 APU was collected from the mechanical department of Nigde Omer Halisdemir University (for chemical composition see Table 1). Powder of Copper Spinel $CuFe_2O_4$ (surface area: 3.6 m$^2$/g and particle size: D50 = 3.3 µm) is an industrial product of fuel cell material and the rest of the chemicals (such as Conc. ethanol, acetone and powder of Poly vinyl butyral) were procured by Malaysian chemical lab Avantis Laboratory Supply (31400 Ipoh, Negeri Perak, Malaysia).

**Table 1.** Chemical composition (wt%) of (a) Crofer 22 APU (b) SUS 430.

|   | Cr | Fe | C | Mn | Si | Cu | Al | S | P | Ti | La | N |
|---|----|----|----|----|----|----|----|----|----|----|----|----|
| A | 22 | Bal. | 0.026 | 0.70 | 0.44 | 0.50 | 0.35 | 0.02 | 0.03 | 0.20 | 0.15 | - |
| B | 17 | - | 0.10 | 1.00 | 0.90 | - | - | 0.03 | 0.04 | - | - | 0.70 |

### 2.2. Powder and Slurry Preparation

For slurry optimization, five samples (A, B, C, D and E) were prepared in unlike ratios of metallic single-phase spinel powder $CuFe_2O_4$ (particle size 3.38 µm) and polymeric binder, i.e., Polyvinyl butyral (PVB) (as shown in Figure 2). The ingredients were mixed together to prepare a blend by using a turbulence mixer for an hour at room temperature. This blend was then dissolved in a solution (i.e., Ethanol) to form slurry which continued to be stirred for about 15 min to avoid formation of beads and to get homogeneity in viscosity. It was observed that increasing the weight percent of metallic powder improved adhesion of the spinel layer, so that the layer did not spall out from the surface. This addition of metallic powder also minimized blister formation. At the same time, ethanol was added to get suitable viscosity of the slurry.

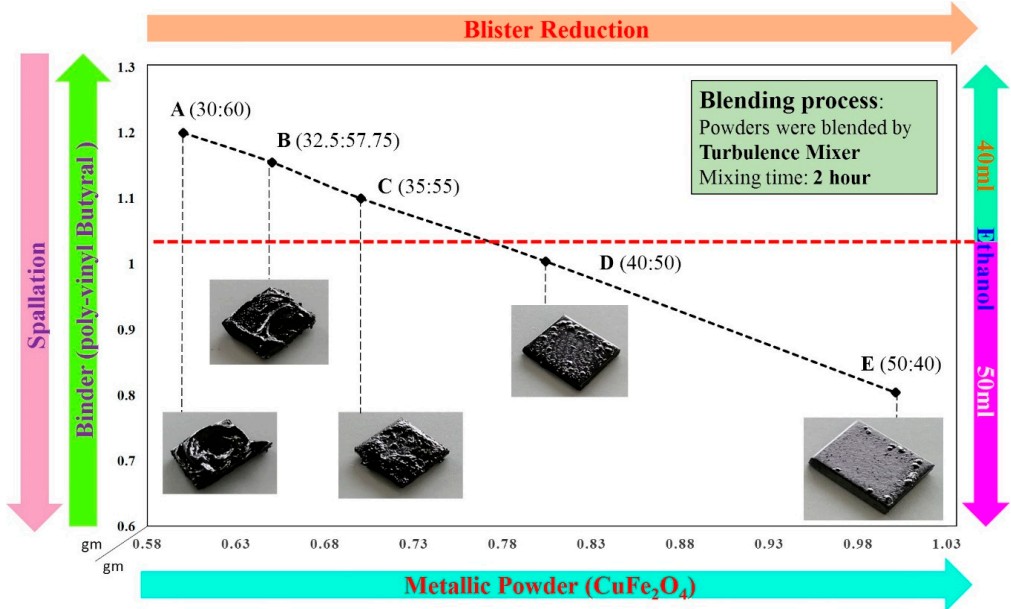

**Figure 2.** Coated sample from different slurry ratios.

## 2.3. Coating and Drying

For the coating, small pieces of Crofer 22 APU and SUS 430 were prepared in a dimension of 20 mm × 20 mm × 2 mm. These pieces were initially grinded by 120 grit and later polished over 1200 grit sandpaper to have a fine and smooth surface. Before coating, these pieces were cleaned in acetone solution and then gently immersed in the prepared slurry. After that samples were hanged for a few seconds for the removal of excess material. For the drying process, coated samples were left in the open air for 4 h and then placed in an oven for 2 h at 70 °C for the removal of volatile species. Heating began with a rate of 0.26 °C/min in a tube furnace where samples were placed in a reduced environment (5% hydrogen and 95% nitrogen) at 800 °C and were soaked for 120 min. After the tube furnace, samples were placed in a box furnace and heated in open-air atmosphere for 2 h at constant temperature and lastly annealed for 6 h in the box furnace (Figure 3 is representing heating and cooling of samples)

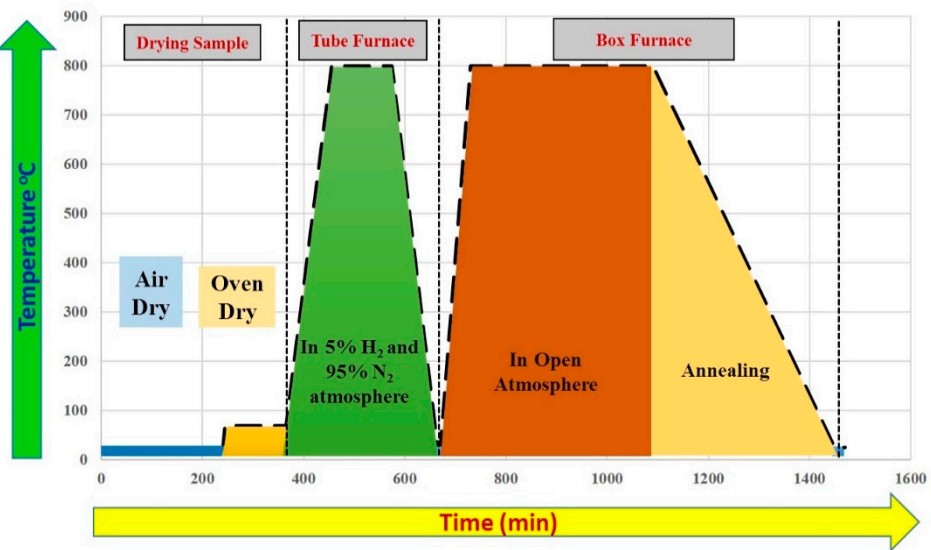

**Figure 3.** Temperature vs. time cycle of annealed samples after coating.

## 2.4. Characterization

For the testing, samples were prepared according to metallography standard (ASTM E3–11). Following the SEM-imaging standard ASTM F1372–93(2020), samples were coated by gold particles to improve conductivity. SEM ZEISS Model: EVO LS15 was used for imaging the cross-sectional area as well as Energy-Dispersive X-ray Spectroscopy (EDX, ZEISS Malaysia). The X-ray Diffraction method was used to analyze phases of spinel layers and for the study of morphological changes of the spinel. To investigate the oxidation behavior of the $CuFe_2O_4$ coated samples, the standard four-probe method was used for ASR test against the function of time. The sample was placed in between Pt-mesh electrodes and Pt-paste was used at the electrode–interconnect interface for better electrical conduction. These electrodes were connected to the source meter Keithley 2400 (Keithley Instruments, LLC, Washington) and then the sample holder was placed in a furnace (setup is shown in Figure 4).

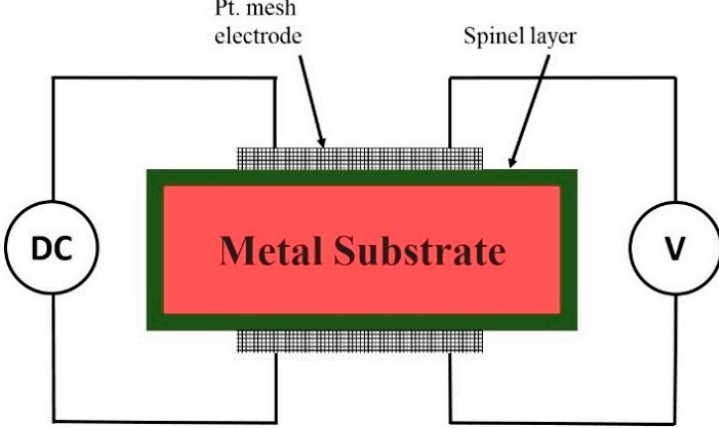

**Figure 4.** Schematic diagram area-specific resistance (ASR) setup.

*2.5. Mathematical Modeling*

Ferritic interconnects commonly form an oxide layer on heating at an elevated temperature. These oxide layers are nonconductive and are formed due to inward diffusion of oxygen. These oxide layers disturb the electrical behavior of materials. Degradation of electrical conduction due to oxides was explained by Adolf Fick. According to Fick's law, the diffusion rate also depends upon the surface area layer, concentration difference and thickness of the oxide scale [33,34]. Another concept for the oxide scale growth was presented by Wagner and termed the Parabolic Rate Law. Wagner's expression for oxide scale thickness is given as [35,36]:

$$\Delta\, x_{ox} \;=\; \sqrt[2]{K_P t} \tag{1}$$

where $x_{ox}$ is the thickness of oxide scale growth, $K_P$ is the weight rate constant and t, time. The above equation (1) shows that the thickness of the scale relies upon the rate formation of the oxide layer and time of oxidation. Formation rate of oxide layer is an intrinsic property which depends upon the chemical composition of the oxide layer. By following the above equation, the thickness of the scale is a direct function of time, meaning that the material will form a thick oxide layer upon exposure in oxidation for a longer period of time. However, an equation of time to produce a specific thickness of an oxide layer ($l_{ox}$) in terms of velocity ($\upsilon_{ox}$) can be written as [37]:

$$t^{lim} \;=\; \frac{l_{ox}}{\upsilon_{ox}} \tag{2}$$

$$K_g \;=\; \frac{1}{t}\left(\frac{\Delta w}{A}\right)^2 \tag{3}$$

where the value of thickness constant ($K_g$) is in terms of weight change ($\Delta w$) and A is the area of oxide scale [31]. The rate constant for the formation of the scale also depends on the temperature of operation. The intense formation of scale increases the thickness of the oxide layer which eventually enhances mass values. Gibbs free energy (heat energy) is necessary to initiate the oxidation reaction which develops the oxide scale. By following the Arrhenius equation of activation energy, the rate formation of $Cr_2O_3$ scale in function activates energy, temperature and weight gain rate constant and can be expressed as [38,39]:

$$K_P \;=\; K_g \cdot e^{\frac{-E_{el}}{R.T}} \tag{4}$$

where $K_g$ is the variable constant that depends upon the rate of chemical reaction, $E_{el}$ is the activation energy for the conduction (75.2 KJ/mol for $Cr_2O_3$), R is the universal gas constant and T is the absolute temperature. Upon realistic conditions of SOFC operation, temperature, activation energy and weight

gain rate constant define the thickness of the oxide scale. For this expression, thickness of oxide on the basis of the above parameters can be written as [40]:

$$\xi^2 = \frac{K_P}{(x\,\rho_{ox})^2}\; e^{\frac{-E_{ox}}{R.T}} \tag{5}$$

$\xi$ is the thickness of the oxide scale ($\Delta\,x_{ox} = \xi$), x is the weight proportion of oxygen and $\rho_{ox}$ (5.22 g·cm$^{-3}$ for $Cr_2O_3$) is the density of the oxide scale and $E_{ox}$ is the activation energy for the growth (220 KJ/mol for $Cr_2O_3$). But the area-specific resistance of the oxide scale depends on the thickness and conductivity ($\sigma_{ox}$) of the oxide scale, which mathematically can be written as [38,41,42]:

$$ASR = \frac{\xi}{\sigma_{ox}} \tag{6}$$

while working in SOFC condition, the scale conduction is not uniform, as it fluctuates with the temperature of the system which lastly affects the electrical resistance of the stack. Therefore, it is necessary to connect the temperature of operation with the conduction of scale, which can mathematically can be written as [43–45]:

$$\sigma_t = \frac{\sigma_{ox}}{T}\; e^{\left(\frac{-E_{el}}{R.T}\right)} \tag{7}$$

here, $\sigma_{ox}$ is the scale conductivity constant ($3.2 \times 10^5$ S/cm for $Cr_2O_3$) and $\sigma_t$ defines the conductivity of scale at a specific temperature. For the purpose of connecting ASR with the weight gain rate, conduction of scale and temperature, Equations (5)–(7) need to be placed in an expression as,

$$ASR^2 = \frac{K_P}{(x\rho_{ox}\sigma_t)^2}\; T^2 e^{\left(\frac{-E_{ox}+2E_{el}}{R.T}\right)} \tag{8}$$

where the equation defines that the ASR value of oxide scale is a function of weight gain rate constant, working temperature and conductance of oxide scale. But the equation is valid only for estimating electrical resistance at a certain point. To study the variation in ASR with respect to time, let it be derived by time operation, which mathematically can be written as:

$$\frac{d(ASR)^2}{dt} = \frac{K_P}{(x\rho_{ox}\sigma_t)^2}\; T^2 e^{\left(\frac{-E_{ox}+2E_{el}}{R.T}\right)} \tag{9}$$

which describes that the growth of the oxide scale depends upon the temperature of condition and the rate formation of the scale. Working at an elevated temperature for a longer period of time increases the thickness of oxide layers, which not only increases the mass value but also increases resistance in the flow of electrons. Temperature also affects electrical conductance of the oxide layer. All these parametric changes influence the area specific resistance of the interconnect material through which the above equation satisfies the experimental conditions of operation.

## 3. Results and Discussion

Crofer 22 APU and SUS 430 are metallic interconnects of SOFC which were coated by a spinel layer of $CuFe_2O_4$ for oxidation resistance. Both steels were coated by dip coating method and annealed to improve performance. The purpose of the spinel layer is to cut off air contact from the chromium oxide layer and to inhibit the formation of the volatile layer $Cr_2(OH)_4$ so that Cr migration can be reduced. For this purpose, the spinel should have better adherence with the substrate and a dense structure that can insulate the base material from the loss of oxidation, electrical and other properties. SEM and XRD (Bruker AXS D8 advance X-Ray Diffractometer, American manufacturer) analysis were used to study the morphological behavior of the material after heat treatment of the samples. EDS line

scanning was used to study the change in alloys concentration, weight gain effect was calculated to check the formation of oxide layers and a 4-probe test was conducted to study the change in electrical conduction during operational hours.

Based on the SEM images of Crofer 22 APU (Figure 5), the spinel layer was successfully fabricated at the surface of Crofer 22 APU. From the imaging technique, it was observed that three different layers were formed after application of the spinel. The X-ray Diffraction pattern identified these layers as the very first layer (L1) being the chromium oxide layer ($Cr_2O_3$), which was formed due high concentration of Cr in substrate. The second (L2) was the spinel protection layer, which was intentionally produced for the reduction in chromium migration, and the third (L3) was the oxide layer of copper (CuO) that was formed during heating in the open environment. Confirming from the results of EDS line scanning, these layers shielded substrate material from mitigation of Cr during formation and operation. Moreover, copper enrichment was also detected at the surface of the base material due to the chemical composition of the spinel.

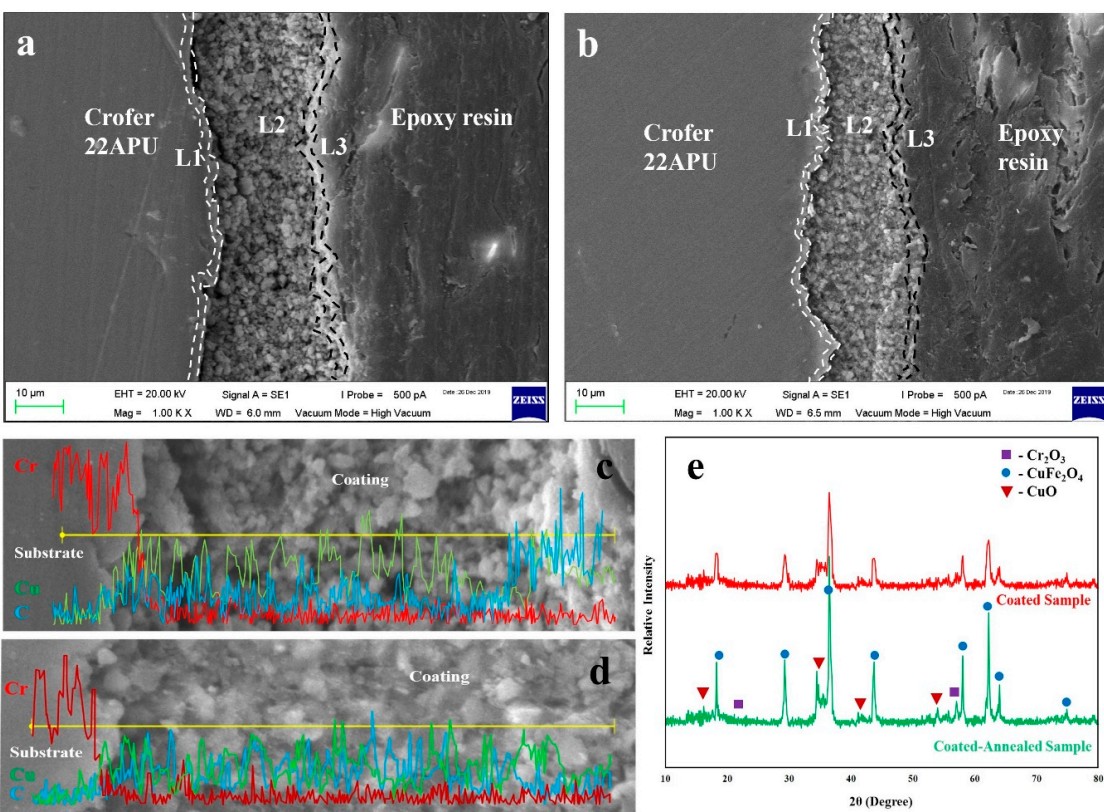

**Figure 5.** Results of Crofer 22 APU before and after annealing of 6 h (**a**,**b**) are SEM-Imaging, (**c**,**d**) are EDS-Line scanning and (**e**) is X-ray diffraction pattern, respectively.

From SEM images of SUS 430 (Figure 6), it is noted that SUS 430 revealed four different layers over the base material upon fabrication of the spinel. The spinel did not spall out during working. These layers were identified by XRD peak pattern as the first (L1) and the last layer (L4) being similar to the Crofer 22 APU that contained chromium oxide ($Cr_2O_3$) and copper oxide, respectively, while the second layer (L2) contained copper, iron, chromium and magnesium oxides due to the interdiffusion of elements. The third layer (L3) was the thin spinel layer, therefore, Cu-spectrum showed inclination in EDS. EDS line scanning confirmed that the coated-SUS 430 successfully decreased mitigation of chromium and showed better performance with the spinel, but described different behavior in the formation of the layer. The trough in line spectrums is the sign of iron and chromium oxide layers under the spinel layer, which is formed due to the internal oxidation of the samples. A special movement of

carbon was noticed in both steels during fabrication of the protection layer. It was observed that the carbon content at the surface was relatively high in all samples.

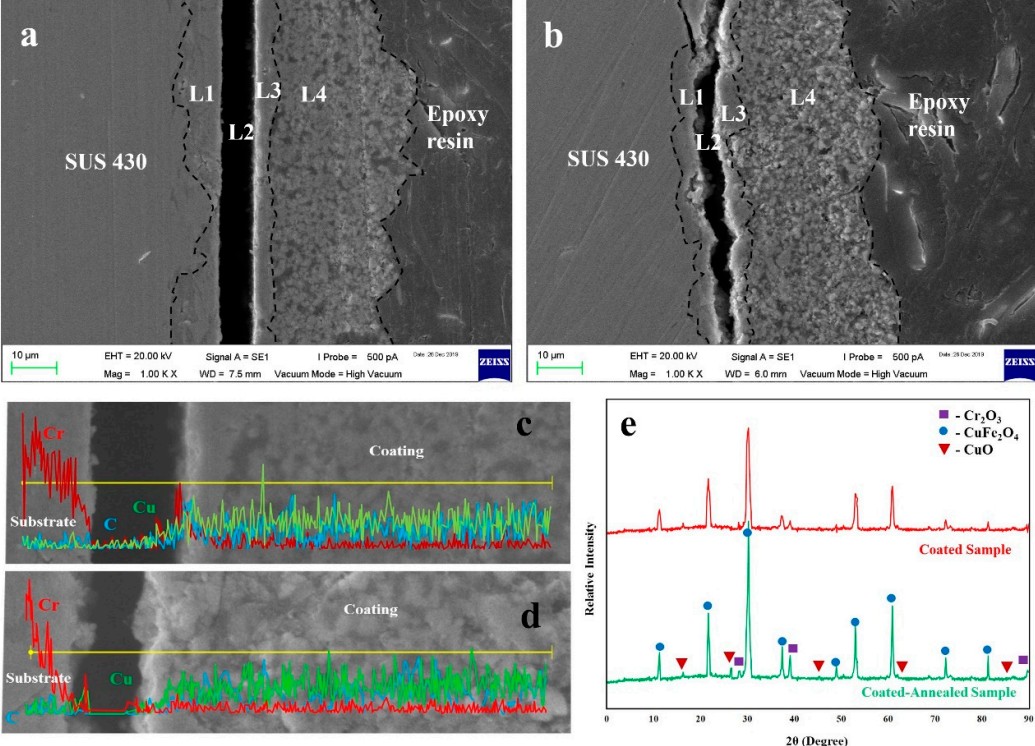

**Figure 6.** Results of SUS 430 before and after annealing of 6 h, (**a**,**b**) are SEM-Imaging, (**c**,**d**) are EDS-Line scanning and (**e**) is X-ray diffraction pattern, respectively.

Heating at 800 °C in the open air increases the mass of both ferritic steels. The results (Figure 7) explain the massive growth of oxide scale over uncoated metallic interconnects. The rapid growth of scale occurred due to the inward diffusion of oxygen at working temperature. The formation of spinel over Crofer 22 APU and SUS 430 reduced growth rate of oxides by $5.3 \times 10^{-14}$ g$^2$·cm$^{-4}$·s$^{-1}$ and $2.1 \times 10^{-13}$ g$^2$·cm$^{-4}$·s$^{-1}$, respectively. The spinel coatings restricted inward diffusion of oxygen which slumped the excessive growth of scale. It was observed that continuous inward diffusion of oxygen increases the thickness of the oxide layer which affects electrical conduction. This oxidation rate was further reduced by annealing process and the weight gain rate for Crofer 22 APU and SUS 430 were calculated as $9.4 \times 10^{-15}$ g$^2$·cm$^{-4}$·s$^{-1}$ and $4.9 \times 10^{-14}$ g$^2$·cm$^{-4}$·s$^{-1}$, respectively.

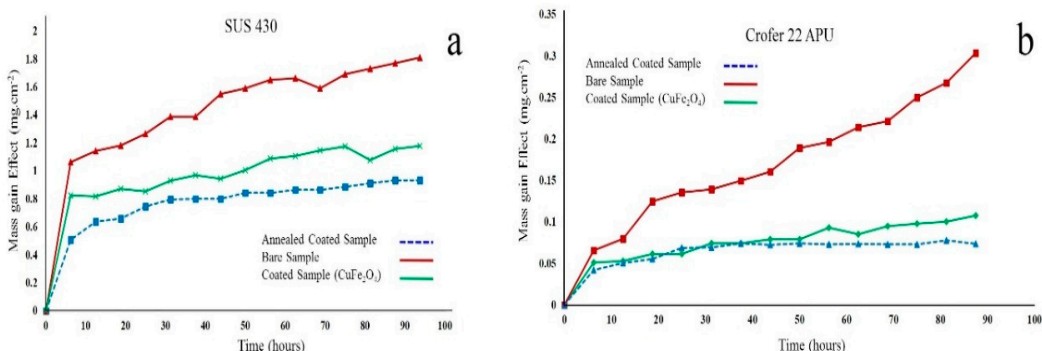

**Figure 7.** Comparisons of coated and annealed with bare samples by parabolic law of weight gain against a time of 12 h (**a**) Crofer 22 APU and (**b**) SUS 430.

Electrical conduction is an important parameter of interconnects. From Figure 8, it was observed that the spinel reduced electrical resistance impressively in both steels. The experimental results showed some fluctuation in readings, but the model data present a linear change in ASR. The reason behind the linear behavior of the model is a parabolic rate constant. From equation 9, ASR value depends on the weight gain of the scale, where $K_p$ is the weight gain rate constant, which shows linear change throughout the operation. On the application of the spinel layer, the material changed its oxidation rate which reduced the scale growth of the steel. At the second stage, annealing changed the structure of spinel and decreased the growth rate of scale which finally hit the parabolic rate constant and assisted the model in a decline of the ASR curve. The experimental values of ASR for annealed samples of Crofer 22 APU and SUS 430 were noted as 10.20 mΩ·cm² and 17.7 mΩ·cm², which are quite similar to the model calculation values: 9.2 mΩ·cm² and 16 mΩ·cm² and much lower than the coated samples, i.e., 17.5 mΩ·cm² and 23.2 mΩ·cm², respectively after oxidation of 400 h at 800 °C.

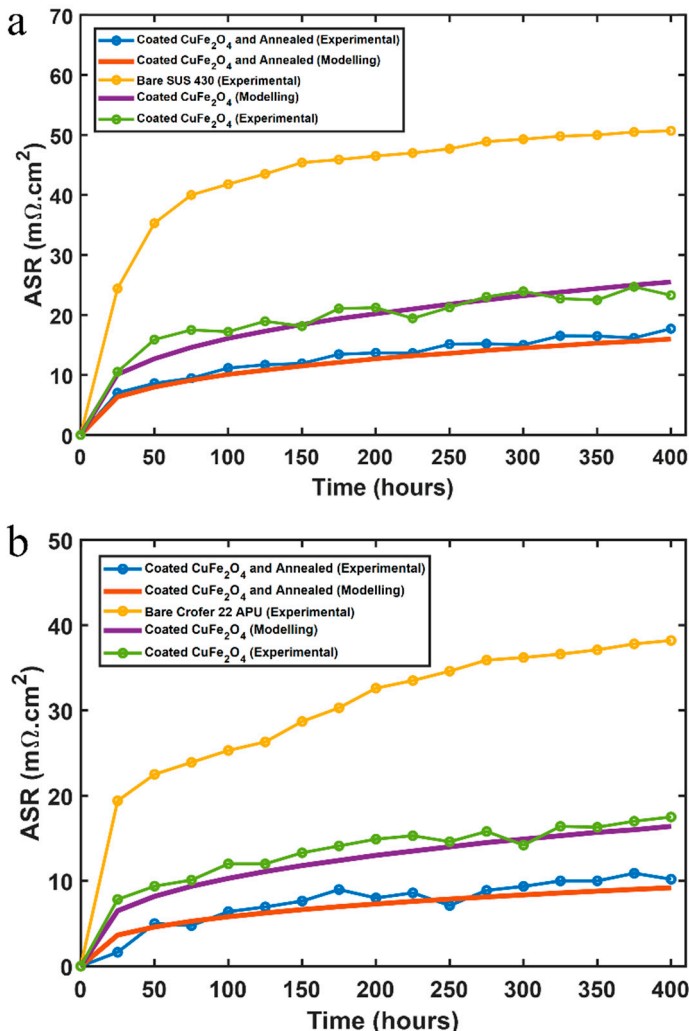

**Figure 8.** Comparison of experimental and modeling data in a coated and annealed sample of (**a**) SUS 430 and (**b**) Crofer 22APU.

The reason behind good adherence with the base material is the presence of iron content. The coherence combination of coating and base material forced them to bind with each other during the intense heating of formation and operation. Based on the SEM-images, the spinel unveiled distinct behaviors with different steels. Crofer 22 APU developed three layers while four layers were formed on the surface of SUS 430. In the case of ferritic steel (SUS 430), the excess formation of the oxide layer (L2) under spinel protection is related to the absence of titanium alloy within the chemical composition.

Alloy addition of titanium opposes internal oxidation during operational heating at 800 °C, therefore, Crofer 22 APU did not produce an extra oxide layer. Nevertheless, the presence of a thick oxide layer did not urge spinel to spall out and did not support Cr to leach out of the interconnect surface during furnace heating.

Due to molecular concentration difference of carbon in substrate and spinel, spinel showed penetration of the carbon during environmental heating (5% hydrogen and 95% nitrogen). The reduced environment permits carbon to gather at the surface of the sample, which is why EDS results showed inclination of carbon at the surface. On heating in the open atmosphere at 800 °C carbon leaches out of the material following the decarburization process. It is the reverse process of carburization in which material was heated in air at an austenizing temperature. As a result, oxygen molecules react with the surface's carbon atoms and produces carbon monoxide ($CO$) and carbon dioxide ($CO_2$) gas which then dissolve into atmosphere [46]. Both steels were heated at the same temperature under a similar environment, but due to alloying difference of carbon within the chemical composition, SUS 430 showed less diffusion of carbon as compared to Crofer 22 APU. According to the iron–carbon diagram, carbon impacts the austenizing temperature of steels, in other words, material with less carbon requires higher temperatures to convert its structure from ferrite into austenite or vice versa [47]. By following $A_3$ line, it is assumed that the SUS 430 showed phase transformation into austenite earlier to Crofer 22 APU which held carbon within the lattice structure of the material and did not allow carbon to diffuse out. In contrast, Crofer 22APU possesses a ferrite structure, which contains less solubility of carbon, thus allowing carbon to diffuse out from grain boundaries. Upon annealing, the spinel layer stopped the carbon diffusing from the substrate but it did not block diffusion from the spinel, and therefore, the line spectrum lowered carbon concentration at the surface on the annealed sample.

High porosity within the spinel layer allows more air to react with the Cr layer which increases the oxidation rate. At the same time, the porous layer increases electrical resistance during operation of the cell because of comparative high thickness and contact resistance of the spinel particles. To overcome these problems, samples were annealed. Annealing allows the carbon to diffuse from the surface of the sample. During heating at 800 °C in open atmosphere, the surface carbon atom reacted with oxygen and formed carbon monoxide and carbon dioxide gas which led to the diffusion of carbon. At the same time, powder particles of the spinel changed their morphological behavior from fine grain to coarse microstructure (as shown in Figure 9). The increasing size of grains reduced the number of grains boundaries, which eventually diminished percent porosity within the lattice. Decreasing porosity of the structure led to densification of the spinel which not only reduced the thickness of the spinel layer but also provided significant resistance in oxidation. On the basis of ASR and mass gain results, densification of the spinel through the annealing process slumps mitigation of chromium and also retards inward diffusion of oxygen. That is why the rate of weight gain reduced. On the other hand, the oxide layers were comparatively thick before heat treatment which defines less density of layers [48].

On combination of SEM and XRD results, it was summarized that annealing did not change the phases of layers but it did reduce the thickness of those layers. The peak of the XRD pattern shows that the density of the spinel increased during the annealing process while the grain size of the spinel was also improved, which led to the reduction in internal porosity of the spinel. Densification of the spinel layer suppressed the oxidation rate of Crofer 22 APU and SUS 430 and also supported the electrical conduction [49–51]. It brought change in density of the spinel which eventually reduced the porosity percentage of the structure, and can be expressed as [48,52]:

$$\%\Phi \;=\; \frac{m}{(A \times h)\phi_p} \tag{10}$$

$\%\Phi$ is percent porosity, m is the mass of a substance, $\phi_p$ is particle density, A (area) and h is the thickness of oxide layer, and particle density is the material constant. Based on the above results, it was suggested that copper-based thin spinel performed better for electrical conduction and that annealing was beneficial for densification of the spinel. The thinning of the spinel layer improved electrical

conduction and copper spinel, not only resolving the chromium leaching problem, but also providing a conductive layer to the interconnect [14,53,54].

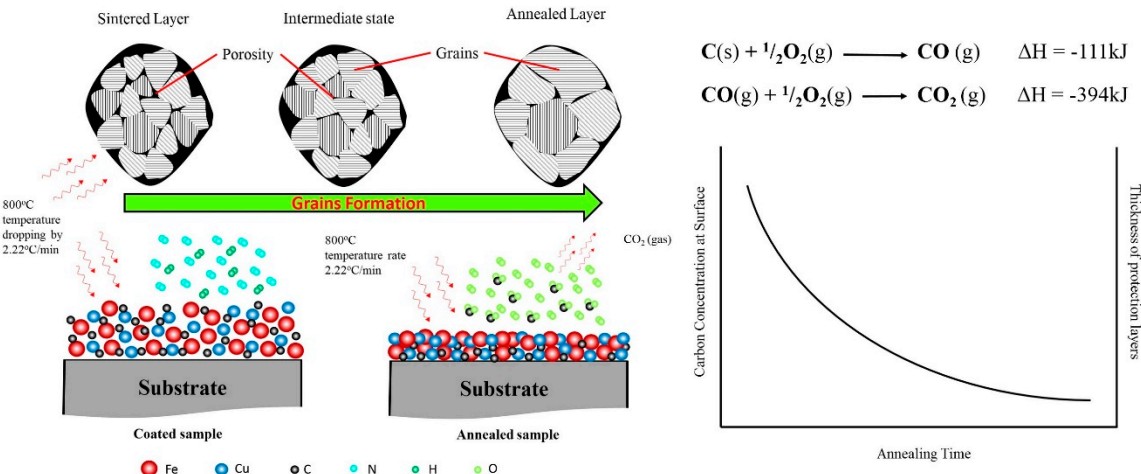

**Figure 9.** Internal mechanism observed during annealing of samples.

## 4. Conclusions

The following conclusion can be derived from this study:

1. The spinel ($CuFe_2O_4$) coated ferritic steel (i.e., SUS 430) is the best combination for interconnects of SOFC due to its good adherence and reduction in oxidation of chromium. The spinel not only diminished the inward diffusion of oxygen but also assisted in electrical conduction of the spinel.
2. The performance of the spinel was improved by the annealing process. Annealing reduced the thickness of the spinel and the crystallinity of oxide layer, which declined the ASR value of the coated SUS 430 by 17.7 mΩ·cm$^2$ after 400 h oxidation at 800 °C.
3. Densification of the spinel layer by annealing proved to be the most promising manufacturing technique. It effectively dropped the percent of porosity in the spinel and decreased the mitigation of chromium significantly. Heat treatment is a satisfactory post-processing method for the production of a thin-dense protective layer.

**Author Contributions:** M.A.H. designed and experimented all the tests and wrote manuscript. O.B.M. assisted in experimentation design, did proof reading of manuscript and supervised this project. All authors have read and agreed to the published version of the manuscript.

**Funding:** The authors would like to thank Yayasan Universiti Teknologi PETRONAS (YUTP) for funding the project under the cost center 015LC0-255.

**Conflicts of Interest:** The authors declare that they have no known competing financial interests or personal relationships that could have appeared to influence the work reported in this paper. This study is part of scope covered in the YUTP 2020 Grant approved on 31 January 2020 under the title of "Enhancement of A Solid Oxide Fuel Cell (SOFC) Interconnect Material Stability by Eliminating the Chromium Leaching via Alloying Addition".

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
