# Peer review of "Mitigation of Chromium Poisoning of Ferritic Interconnect from Annealed Spinel of CuFe2O4"

_processes, doi:10.3390/pr8091113_

Round 1

Reviewer 1 Report

Mitigation of Chromium Poisoning of Ferritic Interconnect from Annealed Spinel of CuFe2O4

The prepared manuscript needs significant revisions, deep discussions and resubmission. In this format I cannot recommend for publication.

1- Introduction section should be strongly modified, and more relevant papers should be added.

2- the discussions are very shallow.

3- Manuscript is not well organized.

4- Figure 2 and 3 need more investigations.

5- More explanations for modeling should be added.

6- A comparative table should be added to compare the results of this work with previous published papers.

7- Fig.9 is the main section and needs deep discussions on the mechanisms. Needs some characterizations.

8- Any fuel cell data?

Reviewer 2 Report

The authors have investigated

“Mitigation of Chromium Poisoning of Ferritic 
 Interconnect from Annealed Spinel of CuFe2O4 
“

The result itself has potential for publication.

1. The study has shed light on how spinel copper ferrous oxide behaves upon annealing. X-ray diffraction study would have added more value to the study.

The authors may also consider including following changes:

2. Line 198 : The figure 6 description is not complete , authors have not included what c ,and d represents .

3. Line 207 : The figure 7 graphs are not legible, authors must improve the quality of the graph

3. Line 210 and 211 : These lines can be improved by rewriting and including  better explanation of the graphs represented in figure 7.

The reviewer has no further comment

Round 2

Reviewer 1 Report

ACCEPT

Well revised.

Reviewer 2 Report

The authors have made the necessary changed asked for .